# Transcription of *WNT* Genes in Hematopoietic Niche’s Mesenchymal Stem Cells in Multiple Myeloma Patients with Different Responses to Treatment

**DOI:** 10.3390/genes14051097

**Published:** 2023-05-17

**Authors:** Natella I. Enukashvily, Liubov. A. Belik, Natalia Yu. Semenova, Ivan I. Kostroma, Ekaterina V. Motyko, Sergey V. Gritsaev, Stanislav S. Bessmeltsev, Sergey V. Sidorkevich, Irina S. Martynkevich

**Affiliations:** 1Russian Research Institute of Hematology and Transfusiology, FMBA of Russian Federation, 191024 St. Petersburg, Russia; 2Institute of Cytology, Russian Academy of Sciences, 194064 St. Petersburg, Russia; 3Cell Technologies Lab., North-Western State Medical University named after I.I. Mechnikov, 191015 St. Peterburg, Russia; 4Cell Technology Center Pokrovsky, 199066 St. Petersburg, Russia

**Keywords:** multiple myeloma, WNT signaling, hematopoietic niche, mesenchymal stromal cells, prognostic markers

## Abstract

Mesenchymal stromal cells (MSCs) are involved in bone tissue remodeling due to their ability to differentiate into osteoblasts and to influence osteoclasts’ activity. Multiple myeloma (MM) is associated with bone resorption. During disease progression, MSCs acquire a tumor-associated phenotype, losing their osteogenic potential. The process is associated with impaired osteoblasts/osteoclasts balance. The WNT signaling pathway plays a major role in maintaining the balance. In MM, it functions in an aberrant way. It is not known yet whether the WNT pathway is restored in patients’ bone narrow after treatment. The aim of the study was to compare the level of *WNT* family gene transcription in the bone marrow MSCs of healthy donors and MM patients before and after therapy. The study included healthy donors (*n* = 3), primary patients (*n* = 3) and patients with different response status to therapy (bortezomib-containing induction regimens) (*n* = 12). The transcription of the *WNT* and *CTNNB1* (encoding β-catenin) genes was accessed using qPCR. The mRNA quantity of ten *WNT* genes, as well as *CTNNB1* mRNA encoding β-catenin, a key mediator in canonical signaling, was evaluated. The observed differences between the groups of patients indicated that aberrant functioning of the WNT pathway was retained after treatment. The differences that we detected for *WNT2B*, *WNT9B* and *CTNNB1* suggested their possible application as prognostic molecular markers.

## 1. Introduction

Multiple myeloma (MM) is a plasma cell clonal proliferative disorder characterized by the abnormal increase in monoclonal immunoglobulins [1]. Cancer plasma cells reside in the bone marrow (BM). In Russia, the annual frequency of new reported cases is approximately 1.9 per 100,000 people. Currently, MM is considered as an incurable disease and causes more than 10,000 deaths each year [2].

The survival of MM tumor cells, their proliferation activity, and the ability to evade the attacks of immune cells and anticancer therapy are facilitated by interactions with elements of the hematopoietic niche (HN), which turns into the tumor microenvironment during tumorigenesis. The stem cell niche is a tissue microenvironment that maintains and regulates stem cell function through cell-to-cell interactions and numerous soluble factors [3,4,5]. HN consists of components of the extracellular matrix and various types of specific, non-hematopoietic, stromal cells. These cells are not involved in hematopoiesis, but they act as a microenvironment for hematopoietic stem cells (HSCs), which give rise to all types of blood cells [6]. Osteoblasts, osteoclasts, endothelial cells, mesenchymal stromal cells (MSCs) and other cells are involved in HN formation. MSCs are the basic component of a healthy HN. They regulate stroma renewal and HSC differentiation [7]. For blood cancers, HN transforms into a new type of niche—a tumor one [8,9,10]. A tumor niche promotes cancer cell proliferation, increases their resistance to therapy and changes the reactions of immune cells. MSCs, after contact with tumor cells, acquire a cancer-associated phenotype (CAPh). MSC with CAPh have many features similar to cancer associated fibroblasts from solid tumors: they increase tumor drug resistance and promote cancer cell proliferation [8,9,10]. At least some cancer-induced changes in HN persist after treatment, although the severity of changes varies in patients with different responses to treatment [11,12].

CAPh MSCs in the cancer HN differ in many aspects from the MSCs in a healthy HN. Many pathways, including the WNT pathway, undergo drastic changes. The pathway is crucial for the regulation of osteoblasto-/osteoclastogenesis, and, thus, for bone tissue formation/resorption. The activation of osteoclasts via RANK/RANKL and Notch pathways leads to bone resorption [13,14]. Bone lesions are common for MM. They occur in 80% of newly diagnosed MM cases and persist even in the absence of active disease, since the complete repair of bone lytic lesions rarely occurs [11,14,15]. Lesions are induced by the impairment in bone formation/regulation balance. The WNT signaling pathway, consisting of WNT family proteins and their receptors, plays an important role in the regulation of the balance, maintaining the osteogenic differentiation of MSCs and preventing osteocyte apoptosis [14,16]. WNT proteins are secreted glycoproteins encoded by 19 genes located on different chromosomes. Inhibition of the pathway decreases osteoblasts’ activity and increases osteoclasts’ activity [17,18]. MM cells secrete WNT antagonists, such as DKK1, blocking the pathway and inducing bone lesions [14].

MSCs of healthy HN secrete WNT ligands such as WNT2, WNT4, WNT5A, WNT11 and WNT16 proteins [19]. Osteoblasts express WNT5A and osteoclasts secrete WNT10B, cross-regulating each other [20,21]. WNT ligands are also expressed by the other cells of the HN. Cancer MM cells secrete a WNT antagonist [22]. These facts suggest that the WNT pathway is deranged in the cancer HN. Since the tumor HN (like any tumor niche) is resistant to antitumor therapy [7,23], it persists during remission and contributes to the occurrence of relapse of the disease. It can be hypothesized that changes in the regulatory processes of osteogenesis, specific to MM, persist during remission in some patients, partly due to disruption of the WNT pathway in the MSCs of the myeloma HN.

The aim of the study was to compare the *WNT* gene expression level in HN MSCs of patients after treatment and healthy BM donors.

## 2. Materials and Methods

### 2.1. Bioethics

All BM samples were obtained from MM patients and healthy BM donors. Sampling was carried out in accordance with the Declaration of Helsinki of the World Medical Association (Declaration of Helsinki: Ethical Principles for Medical Research Involving Human Subjects, including amendments made during the 64th WMA Meeting in Fortaleza, Brazil, October 2013). The study was approved by the Ethics Committee of the Russian Scientific Institute of Hematology and Transfusiology FMBA of Russia (protocol No. 6-2019, 11 June 2019). Written informed consent was obtained from each patient participating in the study.

### 2.2. Patients

Fifteen patients were included in the study after standard bortezomib-based induction therapy was performed in accordance with the national treatment standards, and it included the following regimens: bortezomib and dexamethasone (VD); bortezomib, cyclophosphamide and dexamethasone (VCD); bortezomib, lenalidomide and dexamethasone (VRD) followed by autologous hematopoietic stem cell transplantation (aHSCT) (Table 1). The age of the patients ranged from 49 to 71 years, with a median of 61 years. Treatment efficacy was assessed according to the IMWG (International Myeloma Working Group) criteria (https://www.myeloma.org/resource-library/international-myeloma-working-group-imwg-uniform-response-criteria-multiple, last accessed on 26 December 2022).

The patients were divided into groups, as we described previously [12]. The first group (PoCR) included 9 patients with one of the following response options: complete response (CR), very good partial response (VGPR) and partial response (PR). The second group (NR, *n* = 3) was formed by three patients with no response to the therapy. The control group (HD) included three healthy BM donors. A group of untreated patients (UT, *n* = 3) included three patients with newly diagnosed MM before the beginning of treatment.

### 2.3. Cell Cultures and Lines

BM samples were obtained by iliac crest puncture. The samples were diluted with saline in a ratio of 1:3. Mononuclear cells were isolated according to the standard Ficoll-Paque density gradient isolation protocol (ρ = 1.077 g/cm^3^, PanEco, Russia). A diluted BM sample (7.5 mL) was layered onto 7.5 mL Ficoll-Paque and centrifuged for 40 min at 400 g. The fraction of mononuclear cells was collected, diluted with PBS in a ratio of 1:10 and centrifuged for 10 min at 200 g to remove Ficoll and platelets. The cells were then placed in culture flasks and expanded in low glucose DMEM (Thermofisher, Waltham, MA, USA) supplemented with 10% Fetal Bovine Serum (FBS) (HyClone, Logan, UT, USA), as well as 100 U/mL penicillin and 100 µg/mL streptomycin (Gibco, Thermofisher, Waltham, MA, USA). Cells were incubated at 37 °C in an atmosphere of 5% CO_2_ and 7% O_2_ (conditions of the so-called “physiological hypoxia”, culturing cells at an oxygen concentration corresponding to the tissue one) [24]. The adhesion of MSCs to the plastic occurred after 5–7 days and unattached cells were removed. The medium was changed every 3 days. When a culture density of 70–80% was reached, cells were gathered using solutions of trypsin and versene (Gibco, Thermofisher, Waltham, MA, USA) and subcultured in a ratio of 1:2–1:3. For further experiments, MSCs were used at passages 3–5. The verification of the MSC purity of cell cultures used in the study has been described in detail in [12] (Figure 4, Supplementary Table 1 of the cited work). Briefly, MSC cell cultures met the ISCT criteria with more than 95% of the cells being positive for MSC markers (CD44, CD105, CD90, CD73) and less than 2% being positive for negative MSC markers (CD14, CD45, CD34). The cells were capable of trilineage differentiation, though their osteogenic potential depended on MM progression and response to treatment [12].

### 2.4. Oligonucleotide Sequences: Probes and Primers

The primers published in [25] were used for the amplification of ten genes of the *WNT* family, as well as the β-catenin gene *CTNNB1* (Table 2). Primers for the GAPDH gene (Table 2) were designed during our previous work [12], using the *GAPDH* mRNA reference sequence (accession number in GenBank NM_002046.7) as a template.

### 2.5. RNA Isolation and cDNA Preparation

Total RNA from MSCs was isolated using the ExtractRNA reagent (Evrogen, Moscow, Russia) containing phenol and guanidine isothiocyanate, according to the manufacturer’s protocol. Then, the RNA pellet was dried and dissolved in the RNA buffer included in the RIBO-sorb kit (AmpliSense, Moscow, Russia). The additional purification of RNA from DNA contamination was carried out by incubation with DNase I and EDTA. Then, RNA was stored at −80 °C.

To obtain cDNA from isolated RNA, reverse transcription PCR was performed using M-MuLV-RH reverse transcriptase (Biolabmix, Novosibirsk, Russia). Oligo (dT)_16_ primers were added to the RNA template and incubated for 2 min at 70 °C to melt the secondary structures. Then, buffer and revertase were added and incubated for 60 min at 42 °C. The reaction was stopped by heating to 70 °C for 10 min. The obtained single-stranded cDNA was then quantified using real-time PCR.

### 2.6. Real-Time PCR

Real-time quantitative PCR (qPCR) was performed using the CFX96 Real-Time System (Bio-Rad, Hercules, CA, USA). Samples were prepared using the qPCRmix-HS SYBR qPCR kit (Evrogen, Moscow, Russia).

The following protocol was used for qPCR: 95 °C for 5 min, then 40 cycles of the following steps, 95 °C for 10 s, 56 °C for 30 s and 72 °C for 20 s (3-step protocol). A final heating step of 65 °C to 95 °C was performed to obtain melting curves of final PCR products to check the compliance of the amplification products with the expected ones. mRNA expression levels were calculated by the 2−ΔΔCt method with the gene transcription normalized to a housekeeping gene *GAPDH* encoding glyceraldehyde 3-phosphate dehydrogenase (GAPDH).

### 2.7. Statistical Analysis

Data are representative of three or more independent experiments. qPCR was performed in two biological and three technical replicates for each biological one. GraphPad Prism 8 software was used for plotting and statistical analysis. The results are presented as the mean ± standard error of the mean. The Kruskal–Wallis test was used to analyze the qPCR results. A significant difference was assessed with a *p*-value < 0.05.

## 3. Results

WNT family proteins play a key role in osteogenesis, epithelial–mesenchymal transition, and other processes important for tumor development. In the study, the transcription activity of genes encoding proteins of the WNT family was evaluated in patients before and after MM treatment. Despite the large number of publications about the activity of WNT genes in tumors of various origins, data for multiple myeloma are relatively scarce.

WNT3, WNT3A and WNT5A proteins are known as factors of stimulation (WNT3) or, vice versa, suppression of MM cell growth (WNT3A, WNT5A) [26,27,28,29], but whether their expression level normalizes after treatment is unknown. It is also unknown whether changes affect not only tumor cells but also niche components, especially MSCs, which play an important role in the regulation of the WNT cascade.

In our study, we found significant differences in mRNA level for the *WNT3* gene between MSCs of untreated patients (UT-MSCs) and those obtained from healthy donors (HD-MSCs) (Figure 1A). The result indicated that gene activation in MM occurred not only in tumor cells but also in microenvironment cells, at least in MSCs. The level of the gene transcription in UT-MSCs was increased 165-fold as compared to HD-MSCs. The fold-change value was in the range from 49 to 323 (mean 165.7 ± 80.2) (*p* < 0.05). In most treated patients, the level of *WNT3* transcription returned to normal with the exception of one patient with a partial response, where the level of *WNT3* was 143 times higher than in HD-MSCs. 

MM cells secrete inhibitors of the WNT3A canonical signal transduction pathway, and an increase in WNT3A in MM cells inhibits bone disease and tumor growth in SCID-nu mice [22]. We showed (Figure 1B) that during the disease development, prior to treatment, the level of *WNT3A* mRNA in UT-MSCs increased manyfold (25.4 ± 12.1 vs. HD) in comparison with HD-MSCs. This may be a manifestation of compensatory mechanisms in response to the increased secretion of WNT inhibitors by MM cells. After treatment, regardless of its outcome, the expression level returned to its original level and did not differ from that of HD. It is possible that HN MSCs are involved in compensatory mechanisms.

For WNT5A, there are various data on its involvement in MM. Its expression in MSCs has been previously shown to increase their osteogenic differentiation potential, which is usually impaired in MM [28]. At the same time, the *WNT5A* gene is expressed in the BM of MM patients [27]. According to our data (Figure 1C), the transcription level is increased 2–6-fold (2.9 ± 1.7) in UT-MSCs as compared to HD-MSCs. However, in patients who received treatment but did not respond to it, their gene activity is comparable to that of HDs. Gene activity was very high in the majority of patients with PoCR: the amount of mRNA was 4 to 50 times (42.6 ± 12.0) higher than in the control HD-MSCs. Thus, according to our data, *WNT5A* mRNA quantity increased after treatment, unlike *WNT3A* activity that returned to normal. The observed change in the balance of the two genes of the *WNT* family, i.e., those that are the main regulators of osteogenesis in MM, indicates that complex processes are occurring in the HN in response to signals from MM cells.

WNT5B protein is similar to WNT5A in its composition and functions [30]. However, no data are available on its role in MM progression. Nevertheless, abnormalities in its function are known to cause bone disorders. In the cDNA samples we analyzed (Figure 1D), the *WNT5B* transcription level did not differ from healthy donors. Only in one sample of NR-MSCs (MSCs from a non-responding patient) and in a sample of POCR-, qPCR revealed an increased level of mRNA quantity (MSCs (11.2 ± 1.2 and 27.2 ± 10.7 fold change vs. HD-MSCs, respectively). Thus, despite the similarity in function to WNT5A, the functions of these proteins might be different in MM.

The role of WNT7B protein in the development of MM was reported in a recent study published in 2021 [31], where an increased expression of the *WNT7B* gene promoted tumor growth. We showed that these changes did not affect HN MSCs; the level of WNT7B transcription did not differ between the groups (Figure 1E).

For the WNT8B ligand, there are currently no data on its involvement in the development of MM. However, this protein is involved in the development of some other cancers, such as gastric, breast and other cancers. [32,33]. In our studies (Figure 1F), all MSCs samples, both untreated and treated, had a prominent increase (147- to 4123-fold) in the level of *WNT8B* mRNA as compared to HD-MSCs. 

WNT10B is another member of the WNT protein family. The mRNA level and the role of the protein have not been investigated in MM. However, the protein is known to be involved in the process of osteogenic differentiation as well as in the process of B-cell differentiation [34,35,36,37]. In our study, the level of mRNA in the MSCs of all treated patients was 12– 401 times higher than the *WNT10B* mRNA level in HD-MSCs, and in the MSC of untreated patients, it was 320 times higher as compared to healthy donors’ MSC. No significant differences were found between the groups with different treatment responses; the mRNA level in the cells of treated and non-treated (UT) patients was 147 ± 121 and 320.4 ± 72.1 times higher than the control (Figure 1G). Despite a significant variation in values, an increase in mRNA levels was observed in all MM patients, suggesting a possible role of protein in MM pathogenesis.

WNT2B (earlier known as WNT13) is involved in processes of monocyte differentiation into macrophages [38], suggesting its role in HN formation. We did not detect *WNT2B* mRNA in MSCs in any of the NR patients (Figure 1H), while among UT and PoCR patients, mRNA levels were elevated 9.91–75.45-fold compared to healthy BM donors. It is possible that this increase is related to immune response development. Further studies will answer the question of the prognostic significance of this parameter for the diagnosis and prediction of treatment outcome. mRNA levels of different isoforms of this protein should also be investigated.

WNT9A (formerly WNT14) is reported to be involved in hematopoiesis [39]. It is also expressed in various types of human cancers, such as gastric, pancreatic and breast cancers [40]. In UT-MSCs, *WNT9A* mRNA level was increased in the range from 4.81- to 6072.85-fold in comparison to post-treatment patients and HD (Figure 1I). The significance of this augmentation remains to be investigated.

For the WNT9B protein (formerly WNT15), available data also report its involvement in some cancers, such as breast cancer [41], but not MM. We found that *WNT9B* mRNA was increased 4.7–189.0-fold (vs HD-MSCs) in UT-MSCs and in all MSCs samples obtained from patients who did not respond to treatment (Figure 1J).

β-catenin is an intracellular mediator of the canonical WNT pathway, transmitting signals to transcription factors that activate target gene expression. In MM, it is regulated in two opposite directions: in microenvironment cells, including MSCs, the canonical pathway is suppressed, while in tumor cells, its activation results in an increase in their proliferative activity [42]. For the *CTNNB1* gene encoding β-catenin, we found a significant increase in transcription level in the group of patients responding to therapy (Figure 1K).

The results obtained in the study are summarized in Table 3. The data suggest that some but not all changes in the WNT pathway persist in BM after treatment.

## 4. Discussion

MSCs secrete a number of WNT ligands, such as WNT2, WNT4, WNT5A, WNT11 and WNT16, for interaction with other cells of the microenvironment, as well as for autocrine signaling [19]. These processes activate canonical WNT-signaling, which is based on the accumulation of β-catenin protein in the cell.

In MM cells, an aberrant activity of WNT proteins is observed; increased canonical signaling promotes plasma cell proliferation. The suppression of WNT signals in the microenvironment by tumor-secreted antagonists leads to the suppression of osteogenesis. The mechanisms of WNT dysregulation are now actively being studied. One of the factors affecting signaling activity may be the level of WNT ligands secreted by the cells of both MM and the tumor microenvironment, including MSCs [43]. 

We studied the transcription of *WNT* and *CTNNB1* (β-catenin) genes in MM-MSCs to find out whether their transcription occurs in MSCs of a healthy and tumor HN.

We found an increased transcription of the *WNT3* gene in UT-MSCs. To understand the clinical significance of the persistence of elevated *WNT3* expression in some patients with a partial response, further studies are needed to increase the number of samples analyzed and to compare these with the data of anamnesis and disease outcomes. In a study carried out by Dr. Y. Niitsu’s team, WNT3 was involved in the development of the cell adhesion-mediated drug resistance of MM cells [26]. The tighter adhesion to cells of the BM stroma was accompanied by stronger drug resistance. It was an autocrine effect; a high level of adhesion was associated with an intense expression of *WNT3* in tumor cells. However, this study analyzed the presence of mRNA only in tumor cells, but not in BM cells, and it is not known whether the paracrine transmission of WNT3 to MM cells from MSCs has the same effect.

We also found an increase in *WNT5A* transcription in the PoCR group as compared to the other groups of patients enrolled in the study. WNT5A is known to be a ligand involved in non-canonical signaling. The non-canonical WNT5A/ROR2 pathway was shown to support the osteogenic differentiation of MSCs, which is impaired in MM. The expression of *WNT5A* in MSCs increases the ability of MSCs to differentiate in an osteogenic direction [28]. The increased transcription of *WNT5A* in PoCR patients compared with UT and NR patients, shown in our study, is consistent with this statement; however, the increase in *WNT5A* expression in PoCR patients as compared to HD was an unexpected result. It can be assumed that after successful treatment, *WNT5A* transcription is excessively increased for intensive bone tissue repair. In MM cells, the overexpression of the ROR2 receptor, for which the ligand is WNT5A, led to the formation of cell adhesion-mediated drug resistance [44]. Thus, an increase in *WNT5A* gene transcription in patients of the PoCR group compared to the HD group can be explained by the fact that after therapy, more active metabolic processes occurred in damaged HD, both associated with the restoration of bone tissue and with the expression of signaling molecules and receptors by the MM cells themselves. In a healthy HN, the levels of WNT5A ligands are lower than in MM, since the balance of osteogenesis signaling pathways is not disturbed.

Our studies showed that in all patients, regardless of the treatment outcome, the level of *WNT8B* gene transcription is increased. Despite the fact that *WNT8B* mRNA level and biological functions of the translated protein in MM have not been studied, it is known that the WNT8B protein activates the β-catenin-mediated transition of MSCs to the myofibroblast phenotype [33], which is typical for tumor-associated fibroblasts. We have previously shown that hematopoietic niche MSCs make such a transition when co-cultured with MM cells [12]. Myofibroblasts play a key role in fibrosis development; therefore, *WNT8B* gene activation in the MSCs of patients, both before and after treatment, is most likely associated with the presence of cancer-associated fibroblasts in MSCs cell cultures from the HN of MM patients. The *WNT8B* gene is known to play an important role in the development of idiopathic fibrosis in the lungs [33], along with *WNT10A* and *WNT7B* genes [45]. In some MM patients, HN fibrosis is observed, and it is associated with an increase in plasma cells’ mitotic activity and their lesser differentiation [46]. MSCs and fibroblasts of the solid tumor microenvironment are the main source of extracellular matrix components in fibrosis. However, there are no similar data for HN. In the future, we plan to test whether the presence of the WNT8B ligand in the medium affects the composition and expression level of the extracellular matrix components of HN MSCs and fibroblasts.

The canonical WNT signaling pathway mediated by β-catenin accumulation normally regulates important processes, but in MM it is imbalanced. The canonical pathway in BM is responsible for the balance between osteoblasts and osteoclasts. WNT protein antagonists, released by MM cells, inhibit signaling and impair osteogenesis. At the same time, autocrine effects activate canonical WNT signaling in MM cells, which promotes their proliferation and tumor development [42]. In our study, the transcription of the *CTNNB1* gene encoding β-catenin was increased in PoCR-MSCs compared to both HD-MSCs and UT- and NR-MSCs. Its increase relative to other groups of MM patients—UT and NR—can be interpreted as a result of treatment, which has led to the resumption of the osteogenesis processes in which β-catenin takes part. By analogy with *WNT5A*, it can be assumed that high transcription in the PoCR group is caused by the acceleration of osteogenesis processes.

## 5. Conclusions

The results of our study suggest that WNT pathways are unbalanced in MM patients, and MSCs with a cancer-associated phenotype play an important role in this process. MSCs from MM HN do not fully recover after treatment, both at the level of WNT ligands and their mediators. The cancer-associated phenotype of the MM MSCs of the HN contributes to the MM cells’ drug resistance.

In the HN MSCs of non-responding patients, two genes (*WNT2B*, *WNT9B*) out of those probed in our study were activated in the group of untreated patients with newly diagnosed MM. Of these, the *WNT2B* gene was also active in the group of patients with complete or partial response, but not in the non-responders group. Considering that isoforms of this ligand are involved in the processes of macrophage differentiation and polarization [38], it is possible that its high level in untreated patients is associated with an antitumor response. In contrast, *WNT9B* was active in the non-responder group, but not in the samples of patients with complete or partial response. Previously, it was shown that this gene is not transcribed in the BM of healthy donors. In addition, a significant difference in β-catenin transcription levels was shown for groups of patients.

The results of *WNTs*’ mRNA level assessment in the MSCs of the tumor niche in MM suggest possible prospects for mRNAs of *WNT2B, WNT9B* and *CTNNB1* as being molecular prognostic markers that can predict the outcome of the disease.

## Figures and Tables

**Figure 1 genes-14-01097-f001:**
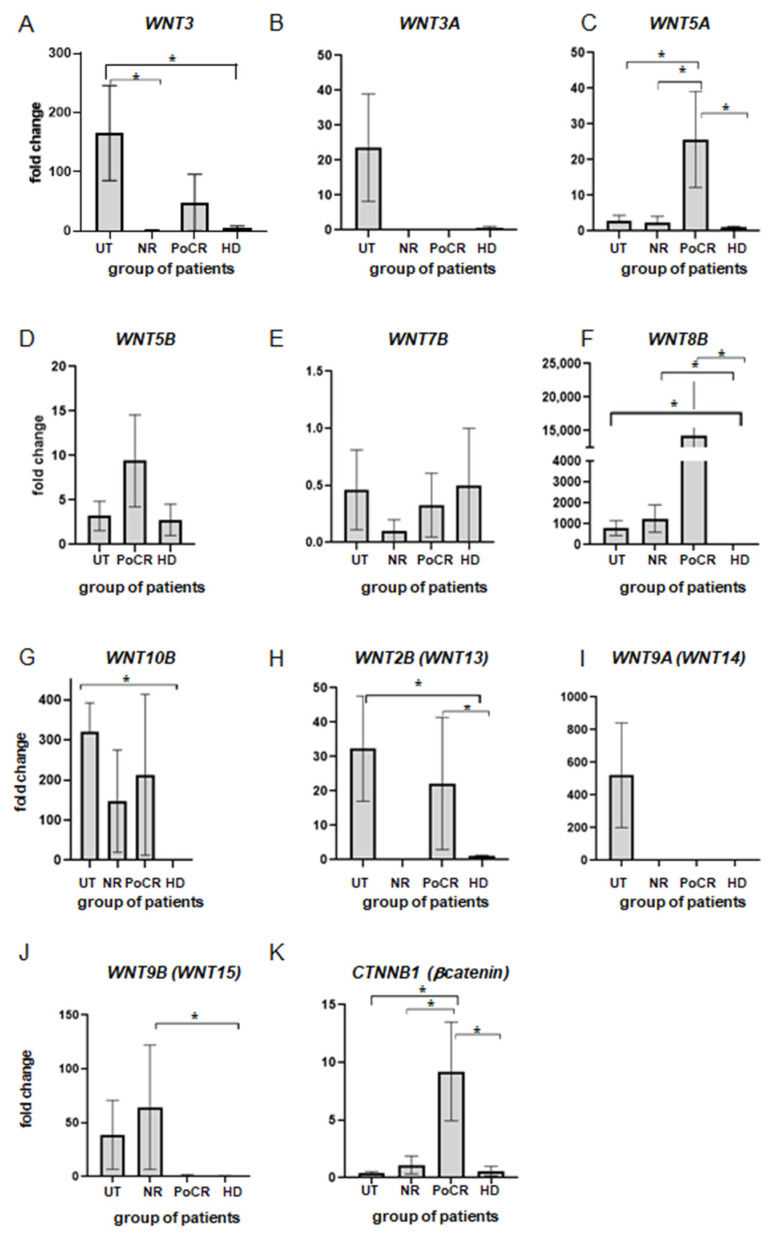
qPCR assessment of *WNT* genes and the *CTNNB1* gene (encoding β-catenin) transcription level (**A**–**K**) in cultures of MSCs of patients with MM; horizontally—groups of patients (UT—untreated patients, *n* = 3; NR—patients without response to the treatment, *n* = 3; PoCR—patients with partial or complete response, *n* = 9; HD—healthy donors, *n* = 3). * *p* < 0.05. The calculation has been carried out for all groups and only significant differences are shown.

**Table 1 genes-14-01097-t001:** Clinical characteristics of patients included in the study.

Sample ID	Type of M Protein	PC Infiltration of BM (Histological Examination of Trephine Biopsy)	Percentage of PC in Bone Marrow Aspirate	Microvessel Density of BM (Histological Examination of Biopsy)	Treatment Regimens *	Response to Therapy
1	Non-secretory MM	10%	6%	7.2%	VCD, 4 cycles	PR
2	Ig G κ	10%	3%	8.9%	VCD, 6 cycles; aHSCT	CR
3	Ig A κ	5%	2.4%	10.2%	VCD, 5 cycles	PR
4	Ig G λ	1–2%	1%	7.4%	VCD, 5 cycles; aHSCT	CR
5	Ig G κ	3%	3.6%	8.5%	VRD, 2 cycles, +2 KRd cycles; aHSCT	VGPR
6	Ig G λ	30%	2.2%	11.7%	VCD, 3 cycles; aHSCT	VGPR
7	Ig G κ	1–2%	1.4%	10.6%	VCD, 5 cycles; aHSCT	PR
8	Ig G κ	1–2%	2.6%	8.6%	CV, 6 cycles	CR
9	Ig G λ	1–2%	2.8%	9.2%	VRD, 1 cycle; aHSCT	VGPR
10	Ig G κ	50%	47%	13%	VD, 6 cycles; aHSCT	NR
11	Ig A κ	50%	14.4%	12.5%	VCD, 3 cycles; aHSCT	NR
12	Ig G λ	90%	82.4%	11.4%	VCD, 2 cycles	NR
13	Ig G κ	N/A	2.4%	N/A	Untreated	UT
14	Ig A	90%	66%	9.3%	Untreated	UT
15	Ig G κ	N/A	84%	N/A	Untreated	UT

* Abbreviations: CR = complete response; aHSCT = autologous hematopoietic stem cell transplantation; PR = partial response; SD = stable disease; VGPR = very good partial response; UT = Untreated; N/A = not available. Regimens abbreviations: CV = bortezomib–cyclophosphamide; VCD = bortezomib–cyclophosphamide–dexamethasone; VD = bortezomib–dexamethasone; VRD = bortezomib–lenalidomide–dexamethasone; KRd = karfilzomib–lenalidomide–dexamethasone.

**Table 2 genes-14-01097-t002:** Primers for detection of cDNA genes of the *WNT* family by real-time PCR.

Gene	Forward Primer	Reverse Primer
*WNT3*	5′-GGAGAAGCGGAAGGAAAAATG-3′	5′-GCACGTCGTAGATGCGAATACA-3′
*WNT3A*	5′-CCTGCACTCCATCCAGCTACA-3′	5′-GACCTCTCTTCCTACCTTTCCCTTA-3′
*WNT5A*	5′-GAAATGCGTGTTGGGTTGAA-3′	5′-ATGCCCTCTCCACAAAGTGAA-3′
*WNT5B*	5′-CTGCCTTTCCAGCGAGAATT-3′	5′-AGGTCAAATGGCCCCCTTT-3′
*WNT7B*	5′-CCCGGCAAGTTCTCTTTCTTC-3′	5′-GGCGTAGCTTTTCTGTGTCCAT-3′
*WNT8B*	5′-TCCCAGAAAAACTGAGGAAACTG-3′	5′-AACCTCTGCCTCTAGGAACCAA-3′
*WNT10B*	5′-CTTTTCAGCCCTTTGCTCTGAT-3′	5′-CCCCTAAAGCTGTTTCCAGGTA-3′
*WNT2B* (former *WNT13*)	5′-TGCCAAGGAGAAGAGGCTTAAG-3′	5′-GTGCGACCACAGCGGTTATT-3′
*WNT9A* (former *WNT14*)	5′-CTTAAGTACAGCAGCAAGTTCGTCAA-3′	5′-CCACGAGGTTGTTGTGGAAGT-3′
*WNT 9B* (former *WNT15*)	5′-CAGGTGCTGAAACTGCGCTAT-3′	5′-GCCCAAGGCCTCATTGGT-3′
*CTNNB1* (*β- catenin*)	5′-CTGCTGTTTTGTTCCGAATGTC-3′	5′-CCATTGGCTCTGTTCTGAAGAGA-3′
*reference gene GAPDH*	5′-AGGTCGGAGTCAACGGATTT-3′	5′-TTCCCGTTCTCAGCCTTGAC-3′

**Table 3 genes-14-01097-t003:** Summary of the *WNT* genes expression level observed in the study.

Gene Name	mRNA Level in MSCs (Relative to HD)		
	UT	NR	PoCR
*WNT3*	+	−	+/−
*WNT3A*	+	−	−
*WNT5A*	+	+/−	+
*WNT5B*	+	+/−	+/−
*WNT7B*	+/−	−	−
*WNT8B*	+	+	+
*WNT10B*	+	+	+
*WNT11*	+	+/−	+/−
*WNT2B (WNT13)*	+/−	−	+/−−
*WNT9A (WNT14)*	+	−	−
*WNT9B (WNT15)*	+	+	+/−
*CTNNB1*	−	+/−	+

+—upregulation, −—downregulation, +/−−—the group includes patients with upregulation and downregulation as compared to HD-MSC.

## Data Availability

Data sharing is not applicable to this article.

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
