# Peer review of "Transcription of WNT Genes in Hematopoietic Niche’s Mesenchymal Stem Cells in Multiple Myeloma Patients with Different Responses to Treatment"

_genes, 2023, doi:10.3390/genes14051097_

Round 1

Reviewer 1 Report

This study aimed to investigate the transcription levels of WNT family genes in bone marrow mesenchymal stem cells (MSCs) from healthy donors and twelve multiple myeloma (MM) patients. The study found significant differences in the transcription levels between the two groups. In particular, the study identified WNT2B, WNT9B, and CTNNB1 as potential prognostic molecular markers for MM.

Critique:

Overall, this is a clearly presented study. However, there are several issues as detailed below that need to be addressed.

Major issues

1.  The WNT family consists of 19 members in human. How about the level of the other 9 genes (WNT1, WNT2, WNT4, WNT6, WNT7A, WNT8A, WNT10A, WNT11 and WNT16) transcription in bone marrow MSCs of healthy donors and MM patients before and after therapy? These genes are also important to MM. 

2. There is no verification of mesenchymal stromal cell purity for these studies. Given that mesenchymal stromal isolation from bone marrow is challenging, this is necessary. See reference (PMID: 17053453) for detail.

3.  English writing details need attention. In the abstract, line 23, β-catenin is a protein that in humans is encoded by the CTNNB1 gene,  β-catenin is not a gene. Line 25, as well as CTNNB1 mRNA encoding β-catenin. Line 26, between (two groups) or among? Line 72, WNT2, WNT4, WNT5A, WNT11 and WNT16. Line 158, GAPDH shoud be italic. Line 178, WNT3 should be in italics. Line 276, CTNNB1 should be in italics. All of the genes should be written in italics, including line 321, line 349, and line 360.

4. In Figure 1, it would be better to conduct a statistical analysis of all of the genes studied. Additionally, the labeling of groups in the figure legend should be clearer, and the placement of labels in the figure should be consistent. WNT3, WNT5A, and CTNNB1 have p values. WNT2B and WNT9B you indicated differences have no p value. And the A, B, C, D, E, and F should be located the same place as the figure. “Group of patients” should be labeled to UT, NR and PoCR. HD means healthy donors (not patient). Why WNT5B and WNT10B do not have NR group??? For UT, NR  PoCR and HD groups, how about the n? This is necessary to write clearly in the figure legend.

Minor issues

1.  In Table 1, the use of "," instead of "." to indicate percentages should be corrected.

2.  In Line 196, and Line 183, it is unclear how the "fold change value range" was determined. It may be more appropriate to report the average values instead.

Author Response

We are very grateful to the reviewer's valuable comments. Our responses are given below. The revised version of the MS with tracked changes is attached to our response.

  1. The WNT family consists of 19 members in human. How about the level of the other 9 genes (WNT1, WNT2, WNT4, WNT6, WNT7A, WNT8A, WNT10A, WNT11 and WNT16) transcription in bone marrow MSCs of healthy donors and MM patients before and after therapy? These genes are also important to MM.

We thank the reviewer for this comment. It is really a very interesting question. However, due to small quantity of cDNA, we do not have enough material for wider screening, the genes of interest were chosen according to the analysis of published data (the references are given in Results and Discussions Sections

  1. There is no verification of mesenchymal stromal cell purity for these studies. Given that mesenchymal stromal isolation from bone marrow is challenging, this is necessary. See reference (PMID: 17053453) for detail.

Thank you for pointing this out. The cell cultures used in the study were characterized in our previous study (https://doi.org/10.3390/ijms23063359, Supplementary Table 1 of the cited study and Figure 4 of the main body). The cell cultures were obtained in Stem Cell Bank Pokrovsky which operated according to the national standards of biobanking. Each cell culture sample is provided with its specification. Besides we accessed surface markers by flow cytometry, trilineage differentiation and the absence of infectious agents during our input controls. We offer our apologies for missing these important data. We have added them into Materials and Methods Section (lines 133-139)

  1. English writing details need attention. In the abstract, line 23, β-catenin is a protein that in humans is encoded by the CTNNB1 gene, β-catenin is not a gene. Line 25, as well as CTNNB1 mRNA encoding β-catenin. Line 26, between (two groups) or among? Line 72, WNT2, WNT4, WNT5A, WNT11 and WNT16. Line 158, GAPDH shoud be italic. Line 178, WNT3 should be in italics. Line 276, CTNNB1 should be in italics. All of the genes should be written in italics, including line 321, line 349, and line 360.

We are very grateful to the reviewer for this comment. We have corrected the text: either type a gene in Italic or correct the text if it meant proteins

  1. 4. In Figure 1, it would be better to conduct a statistical analysis of all of the genes studied. Additionally, the labeling of groups in the figure legend should be clearer, and the placement of labels in the figure should be consistent. WNT3, WNT5A, and CTNNB1 have p values. WNT2B and WNT9B you indicated differences have no p value. And the A, B, C, D, E, and F should be located the same place as the figure. “Group of patients” should be labeled to UT, NR and PoCR. HD means healthy donors (not patient). Why WNT5B and WNT10B do not have NR group??? For UT, NR PoCR and HD groups, how about the n? This is necessary to write clearly in the figure legend.

The Figure has been revised according to the reviewer’s comment: UT value has been added to WNT5B and WNT10B (the MS text has been also revised, lines 266-267), number of patients or donors is specified in the legend now. Statistical calculations were carried out for all the genes, but only significant difference is labeled. We have corrected the figure to make it clear

Minor issues

  1. In Table 1, the use of "," instead of "." to indicate percentages should be corrected.

This minor issue has been corrected

  1. In Line 196, and Line 183, it is unclear how the "fold change value range" was determined. It may be more appropriate to report the average values instead.

We have corrected the issue (lines 195, 209)

Reviewer 2 Report

Interesting study with some concerns:

Nowadays, daratumumab is being used in the frontline or subsequent lines of treatment and analyzing all the pathways would be more comprehensive if these subsets of patients are included. 

The finding presented are likely for patients treated in frontline but how about R/R setting. 

The results/discussion are hard to follow.Recommend in the results section to avoid excessive explanation/interpretation and add this in the discussion. 

Wnt3 is increased in myeloma patients however this does not imply a causation. In line 188, I would rephrase that it is involved in the development of MM and just indicate that its level is increased in MM. Alternatively, use less assertive language especially that the study involved only 12 patients. You next mention that this could be compensatory. 

Include a table summarizing your results highlighting the differences in newly diagnosed, treated and healthy individuals.

Line 48:Needs reference "In blood cancers, 48 HN is transformed into a new type of niche – a tumor one."

Line 78; you mean the pathway is deranged not rearranged.

In the characteristics table, what do you mean by myelogram? Unclear. 

Line 115: Is it sternum or iliac crest? 

Some of the language is incoherent as a result of translation. 

Author Response

We are very grateful to Reviewer2 for helpful comments which allowed us to improve the manuscript. Our responses are given below. A PDF-file of the revised version with tracked changes is attached to the response.

  1. 1. Nowadays, daratumumab is being used in the frontline or subsequent lines of treatment and analyzing all the pathways would be more comprehensive if these subsets of patients are included.

Daratumumab is registered and included in protocols  in our country only for frontline treatment of patients  that are not suitable for autoHCT.  For those patients who are candidates for autoHCT, the treatment is not registered. Moreover, it is preferentially used for treatment of patients with cytogenetically abnormalities. Therefore we do not have samples from the patients with this treatment. However, we are grateful to the reviewer  for this valuable comment. It gave us some ideas for future in vitro experiments with cells from untreated patients

The finding presented are likely for patients treated in frontline but how about R/R setting.

Some of the patients of non-responders group are now candidates for second line therapy, we plan to obtain samples of BM MSC from them to continue our study. But now, we do not have these results.

The results/discussion are hard to follow. Recommend in the results section to avoid excessive explanation/interpretation and add this in the discussion.

The text has been revised according to the reviewer’s suggestion.

Wnt3 is increased in myeloma patients however this does not imply a causation. In line 188, I would rephrase that it is involved in the development of MM and just indicate that its level is increased in MM. Alternatively, use less assertive language especially that the study involved only 12 patients. You next mention that this could be compensatory.

The text has been revised according to the reviewer’s suggestion (lines 199-204 in the revised version are deleted text. Instead, lines 314-317 has been inserted in Discussion section)

Include a table summarizing your results highlighting the differences in newly diagnosed, treated and healthy individuals.

We are grateful for this recommendation. The table is now included as Table 3 (line 296 and below). The text of the MS has been revised accordingly (lines 299-300)

Line 48:Needs reference "In blood cancers, 48 HN is transformed into a new type of niche – a tumor one."

We have added the references (line 51)

Line 78; you mean the pathway is deranged not rearranged.

The mistake has been corrected (line 76). Thank you for pointing this out.

In the characteristics table, what do you mean by myelogram? Unclear.

We offer our apology for the translation mistake. The term ‘myelogram’ in medical vocabulary in our language means ‘qualitative and quantitative study of the cellular composition of the bone marrow obtained by puncture’ . When calculating the myelogram, the cellularity of the composition, the appearance of the cells (mono / polymorphism), the number of megakaryocytes, the presence of clusters of tumor cells are determined.  All vocabulars translate the term into English as ‘myelogram’

In English medical language, as we can see now, the meaning is totally different –  a myelogram is a diagnostic imaging test generally done by a radiologist.

In Table 1, we present two ways of PC calculation in patients’ bone marrow –in bone marrow trephine biopsy (in non-revised Table 1 – PC infiltration of BM according to histology) and in bone marrow  aspirate (Myelogram in non-revised version of Table 1)

We are very grateful for drawing our attention to this mistake. Table 1 has been corrected in the revised version (lines  111).

Line 115: Is it sternum or iliac crest?

The sentence has been corrected. We checked patients’ clinical records. All samples included in the study were obtained by iliac crest puncture. (line 118)

Some of the language is incoherent as a result of translation.

The language correction has been carried out

Round 2

Reviewer 1 Report

Thank you for addressing my concerns and revising the manuscript. However, there are still several major issues that need to be addressed:

1. The abstract states that the study included n=12 participants, including healthy donors, primary patients, and patients with different response statuses to therapy. However, Table 1 lists n=14 sample IDs. In line 95, it is written that twelve patients were included in the study. As the number of patients is a critical aspect of the study, it should be clearly and consistently stated throughout the manuscript.

2. The number of samples for each group in Figure 1 is unclear. The number of samples for untreated patients is only n=2, which is not sufficient for drawing biologically meaningful conclusions. Additionally, the number of healthy donors is not listed. The figure should clearly indicate the number of samples in each group to facilitate understanding.

3. Figure 1C still has an incorrect label. It is unclear which two samples are being compared as the third line. This should be corrected to avoid confusion.

Thank you for taking the time to address these issues. Clarifying these points will improve the clarity and accuracy of the manuscript.

Author Response

We are very grateful to the reviewer for very useful comments. Our answer to them are given below.

  1. The abstract states that the study included n=12 participants, including healthy donors, primary patients, and patients with different response statuses to therapy. However, Table 1 lists n=14 sample IDs. In line 95, it is written that twelve patients were included in the study. As the number of patients is a critical aspect of the study, it should be clearly and consistently stated throughout the manuscript.

Thank you for pointing this out. The study now includes 3 donors and 15 patients (please see our answer to comment 2, in the previous version of the manuscript, the study includes 14 patients and 3 healthy donors). Currently, the number of samples is shown in lines 21-23, 106-110, 230-234

  1. The number of samples for each group in Figure 1 is unclear. The number of samples for untreated patients is only n=2, which is not sufficient for drawing biologically meaningful conclusions. Additionally, the number of healthy donors is not listed. The figure should clearly indicate the number of samples in each group to facilitate understanding.

We were lucky to get another untreated sample during round 2 of revision. This is the reason for our delayed resubmission. It is included now in the table 1, the values for  WNT transcription in UT group  have been corrected where appropriate  .  However, the trends and statistical calculations were not significantly influenced with the addition of the third UT sample. Now, untreated group consists of three patients, non-responders group includes also 3 patients, responders (POCR) – 9 patients. HD group consists also of three samples. Therefore the total number of samples is 18 (including 15 patients).

  1. Figure 1C still has an incorrect label. It is unclear which two samples are being compared as the third line. This should be corrected to avoid confusion.

The Figure has been corrected.

Round 3

Reviewer 1 Report

The author's response to my previous comments. After reviewing the revised manuscript and the author's response, I am pleased to inform you that I am fine with the revision.